# Nuclear dengue virus NS5 antagonizes expression of PAF1-dependent immune response genes

Marine J. Petit[1,2¤a], Matthew W. Kenaston[1], Oanh H. Pham[1], Ariana A. Nagainis[1,2¤b], Adam T. Fishburn[1], Priya S. Shah[1,2]*

**1** Department of Microbiology and Molecular Genetics, University of California, Davis, California, United States of America, **2** Department of Chemical Engineering, University of California, Davis, California, United States of America

¤a Current address: University of Glasgow MRC-Centre for Virus Research, Glasgow, United Kingdom
¤b Current address: San Francisco State University, San Francisco, California, United States of America
* prsshah@ucdavis.edu

**Data Availability Statement:** All transcriptomics data are available on NCBI BioProject database. Accession number are PRJNA720279 and PRJNA720413.

## Abstract

Dengue virus (DENV) disruption of the innate immune response is critical to establish infection. DENV non-structural protein 5 (NS5) plays a central role in this disruption, such as antagonism of STAT2. We recently found that DENV serotype 2 (DENV2) NS5 interacts with Polymerase associated factor 1 complex (PAF1C). The primary members of PAF1C are PAF1, LEO1, CTR9, and CDC73. This nuclear complex is an emerging player in the immune response. It promotes the expression of many genes, including genes related to the antiviral, antimicrobial and inflammatory responses, through close association with the chromatin of these genes. Our previous work demonstrated that NS5 antagonizes PAF1C recruitment to immune response genes. However, it remains unknown if NS5 antagonism of PAF1C is complementary to its antagonism of STAT2. Here, we show that knockout of PAF1 enhances DENV2 infectious virion production. By comparing gene expression profiles in PAF1 and STAT2 knockout cells, we find that PAF1 is necessary to express immune response genes that are STAT2-independent. Finally, we mapped the viral determinants for the NS5-PAF1C protein interaction. We found that NS5 nuclear localization and the C-terminal region of the methyltransferase domain are required for its interaction with PAF1C. Mutation of these regions rescued the expression of PAF1-dependent immune response genes that are antagonized by NS5. In sum, our results support a role for PAF1C in restricting DENV2 replication that NS5 antagonizes through its protein interaction with PAF1C.

## Author summary

Dengue virus (DENV) is a pathogen that infects nearly 400 million people a year and thus represents a major challenge for public health. Productive infection by DENV relies on the effective evasion of intrinsic antiviral defenses and is often accomplished through virus-host protein interactions. Here, we investigate the recently discovered interaction

**Funding:** Funding was provided by University of California, Davis and the W. M. Keck foundation to PSS. MJP was partially supported by the Philippe Foundation Inc. MWK was partially supported by the UC Davis Provost's Undergraduate Fellowship. ATF was supported by a NIH T32 fellowship (2T32AI060555-16). The sequencing was carried out at the DNA Technologies and Expression Analysis Cores at the UC Davis Genome Center, supported by NIH Shared Instrumentation Grant 1S10OD010786-01. The Olympus FV1000 confocal used in this study was purchased using NIH Shared Instrumentation Grant 1S10RR019266-01. The funders had no role in study design, data collection and analysis, decision to publish, or preparation of the manuscript.

**Competing interests:** The authors have declared that no competing interests exist.

between DENV non-structural protein 5 (NS5) and the transcriptional regulator Polymerase associated factor 1 complex (PAF1C). Our work demonstrates PAF1C member PAF1 acts as an antiviral factor and inhibits DENV replication. In parallel, we identified immune response genes involved in intrinsic antiviral defense that depend on PAF1 for expression. We further identified the regions of NS5 required for the protein interaction with PAF1C. Breaking the NS5-PAF1C protein interaction restores the expression of PAF1-dependent immune response genes. Together, our work establishes the antiviral role of PAF1C in DENV infection and NS5 antagonism of PAF1-dependent gene expression through a virus-host protein interaction.

## Introduction

Dengue virus (DENV) is a major source of human disease and is responsible for nearly 400 million infections annually [1]. DENV belongs to the *Flavivirus* genus of viruses, which are positive sense, capped single-stranded RNA viruses that rely on multifunctional proteins to replicate. Flavivirus non-structural protein 5 (NS5) functions as both the RNA-dependent RNA polymerase (RdRp) [2] and the methyltransferase (MTase) domain, displaying N7 and 2'-O methylation [3–5]. Genome replication is the main enzymatic activity of NS5 and occurs on the cytosolic side of ER-associated viral replication factories. In addition to playing a central role in genome replication, NS5 is a key player in disabling the innate immune response during DENV infection. Importantly, host protein interactions with NS5 are critical to its role in immune evasion. DENV NS5 binds human STAT2 and targets it for ubiquitin-mediated proteasomal degradation. This inhibits the expression of interferon-stimulated genes (ISGs) through interferon type 1 (IFN-I) signaling [6].

Interestingly, NS5 also has a nuclear role. DENV serotype 2 (DENV2) NS5 steady-state localization to the nucleus during infection has been known for over 25 years [7] and observed by several other groups independently [8–11]. For other DENV serotypes, NS5 has shown varying steady-state distribution between the nucleus and cytoplasm. For example, DENV1 NS5 is equally distributed across the cell while DENV2 NS5 is predominantly nuclear [9,11]. Despite these variations in NS5 nuclear localization, the consistent detection of NS5 in the nucleus suggests the conservation of NS5 nuclear-cytoplasmic shuttle in mammalian cells. DENV2 mutants that retain some NS5 nuclear localization are viable and can even replicate at near-wild-type (WT) levels. On the other hand, mutants that essentially eliminate NS5 nuclear localization are lethal [8–10]. These defects in replication are not due to decreased intrinsic enzyme activity of NS5 [8,9]. We and others have shown that NS5 of DENV2 16681 interacts with many nuclear proteins [12–14] and inhibits antiviral gene expression, potentially through these interactions [13,14]. Thus, some nuclear localization of DENV2 16681 NS5 is essential for replication and rewiring host gene expression, but the mechanisms by which nuclear DENV2 NS5 accomplishes these tasks are not completely understood.

Using a comprehensive global proteomics approach that defined the DENV-host protein interaction landscape, we recently found DENV2 16681 NS5 interacts with and antagonizes Polymerase associated factor 1 complex (PAF1C) [14]. This nuclear complex, whose primary members are PAF1, LEO1, CTR9, and CDC73, has emerging significance in immunology. The complex generally promotes the expression of many genes, including genes related to the antiviral, antimicrobial, and inflammatory responses [15,16]. We previously showed that NS5 antagonizes PAF1C by inhibiting its recruitment to immune response genes [14]. However, it is currently unknown if NS5 inhibits expression of immune response genes distinct from

STAT2-dependent ISGs through this protein interaction. Such antagonism may also explain a possible synergistic role for nuclear NS5. Yet, these mechanistic details have not been explored.

Here, we investigate if NS5 antagonism of PAF1C may be complementary to the antagonism of STAT2. We find that PAF1 restricts DENV2 infectious virion production. We also find that PAF1 is required for the expression of genes that are primarily STAT2-independent following innate immune stimulation. We further map the viral determinants of the NS5-PAF1C interaction using affinity purification and immunoblot of NS5 mutants to show that NS5 nuclear localization and the C-terminal region of the MTase domain are required for its interaction with PAF1C. NS5 mutants with reduced binding to PAF1C also have reduced antagonism of PAF1-dependent and STAT2-independent genes. Taken together, PAF1C-mediated expression of immune response genes is STAT2-independent and breaking the NS5-PAF1C interaction rescues PAF1-dependent gene expression.

## Results

### PAF1C member PAF1 restricts DENV2 replication

PAF1C is an emergent antiviral factor in mammals [15,16]. We previously showed that PAF1C restricts DENV2 replication by quantifying DENV antigen-positive cells by immuno-fluorescence microscopy [14]. To determine the impact of PAF1C on the full viral replication cycle, we measured infectious virion production in PAF1 knockout (PAF1 KO) A549 cells at several time points. Since the entire complex can be destabilized with depletion of PAF1 only [17], we chose this member as a target for knockout. Generating PAF1 KO cells required isolating single clones. Therefore, we generated PAF1 rescue cells to control for clonal and CRISPR/Cas9 off-target effects (Figs 1A and S1A). Infectious virion production increased approximately 3-fold at 48- and 72-hours post-infection in PAF1 KO cells (Fig 1B). This replication advantage in PAF1 KO cells was significant ($p \leq 0.05$) but disappeared at 96 hours post-infection. To account for the two rounds of lentiviral transduction the PAF1 rescue cells experienced, we created PAF1 KO cells transduced with a GFP-expressing lentivirus. We also created a non-targeting gRNA from parental A549 cells, though this cell line is not clonally selected like the PAF1 KO cells. While overall replication kinetics were slightly different in this experiment, overall trends remained the same. Infectious virion production was approximately

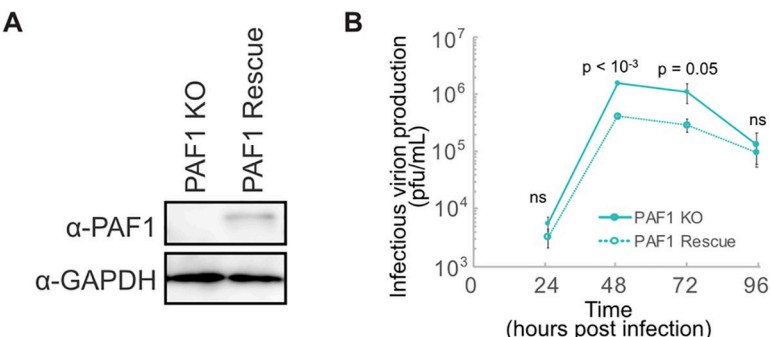

**Fig 1. PAF1 restricts DENV2 infectious virion production.** (A) Immunoblot analysis of PAF1 expression in PAF1 KO and rescue A549 cells. GAPDH is a loading control. (B) DENV2 replication in PAF1 KO and rescue cells, MOI 0.1. Data from three independent biological replicates are plotted as mean values +/- standard deviation. P values were calculated using a paired, one-tailed Student's t-test. Abbreviations: plaque forming units (pfu), not statistically significant (ns).

5- and 14-fold higher at 72 and 96 hours post-infection, respectively in GFP-expressing PAF1 KO cells compared to PAF1 rescue cells (S1C Fig). The replication advantage in PAF1 KO cells was significant ($p < 0.05$ and $p < 0.005$). The cell line with non-targeting gRNA did display different replication kinetics, likely due to differences arising from clonal selection of the PAF1 KO cells, which were used to generate the PAF1 rescue cells (S1D Fig). An immunoblot comparing PAF1 expression in the cell lysate of the PAF1 rescue cells and non-targeting gRNA cell line shows lower expression in the PAF1 rescue cells (S1A Fig). Immunofluorescence microscopy analysis revealed slightly higher fluorescence intensity in PAF1 rescue cells, possibly resulting from a different antibody used for immunofluorescence compared to immunoblot. Notably, this difference in intensity was not significant ($p = 0.98$), and the heterogeneity of PAF1 expression in rescue cells was similar to parental cells (S1E and S1F Fig). These results suggest that the double transduction does not affect the overall behavior of PAF1 rescue cells compared to PAF1 KO cells. Based on these results, we conclude that PAF1 restricts DENV2 infectious virion production in human cells.

## PAF1 is required for the expression of STAT2-independent genes

PAF1C regulates the expression of stress response genes [14], including ISGs and inflammatory genes [15,16]. Since NS5 interacts with STAT2 and PAF1C, we hypothesized that PAF1C could play a unique role upstream or independent of STAT2-mediated IFN-I signaling. For this reason, we studied gene expression following activation of the immune response using a three-hour treatment with poly(I:C), a dsRNA mimic that can stimulate RIG-I, MDA5 and TLR3 signaling [18–21]. Importantly, our treatment was long enough to induce IFNBI expression in A549 cells (log2 fold change 11.9, p adj $< 10^{-19}$) (S1 Table) and could thus capture events upstream and downstream of IFN-I signaling. To distinguish the PAF1- and STAT2-dependent responses, we performed experiments in PAF1 and STAT2 KO and rescue cells (Fig 2A). We assessed data quality on a global level using principal component analysis (PCA) (Fig 2B). Strong clustering of parental A549, PAF1 rescue, and STAT2 rescue cells suggests the effects observed on gene expression are not a result of clonal or CRISPR/Cas9 off-target effects, and we can compare the different KO cells to the same parental control for gene expression. It also suggests that the replication difference between non-targeting gRNA cells (bulk population) and PAF1 rescue cells (created from a clonal PAF1 KO line) (S1C Fig) is driven by differences not captured in the PCA, such as changes in cell metabolism or survival. Additionally, we show that PAF1 and STAT2 KO form distinct clusters for both poly(I:C) treated and control samples, suggesting that PAF1 and STAT2 have unique roles in basal and immune gene expression.

To gain additional insight into PAF1- and STAT2-mediated gene expression, we analyzed global changes using Gene Set Enrichment Analysis (GSEA) [22,23]. Several Reactome pathways were significantly enriched among PAF1 and STAT2 KO cells compared to parental A549 cells (Fig 2C and S1 and S2 Tables). Distinct enriched pathways were observed for each genotype, corroborating our PCA results. As expected, pathways related to IFN-I signaling were significantly downregulated in STAT2 KO cells (p adj $< 0.01$). Extracellular matrix organization (ECM) pathways were also significantly upregulated in STAT2 KO cells, indicating an additional role for STAT2 in repressing the expression of these genes (p adj $< 0.01$). Top downregulated categories in PAF1 KO cells were related to PAF1C roles in chromatin modification (p adj $< 0.005$) or regulation of TP53 and DNA double-strand break repair pathways (p adj $< 0.1$). One significantly downregulated category, the DDX58-IFIH1 (RIG-I-MDA5) induction of IFN-I response (p adj $< 0.05$), is important for the production of IFN-I during DENV infection [24–29]. Therefore, PAF1 may regulate the immune response upstream of IFN-I

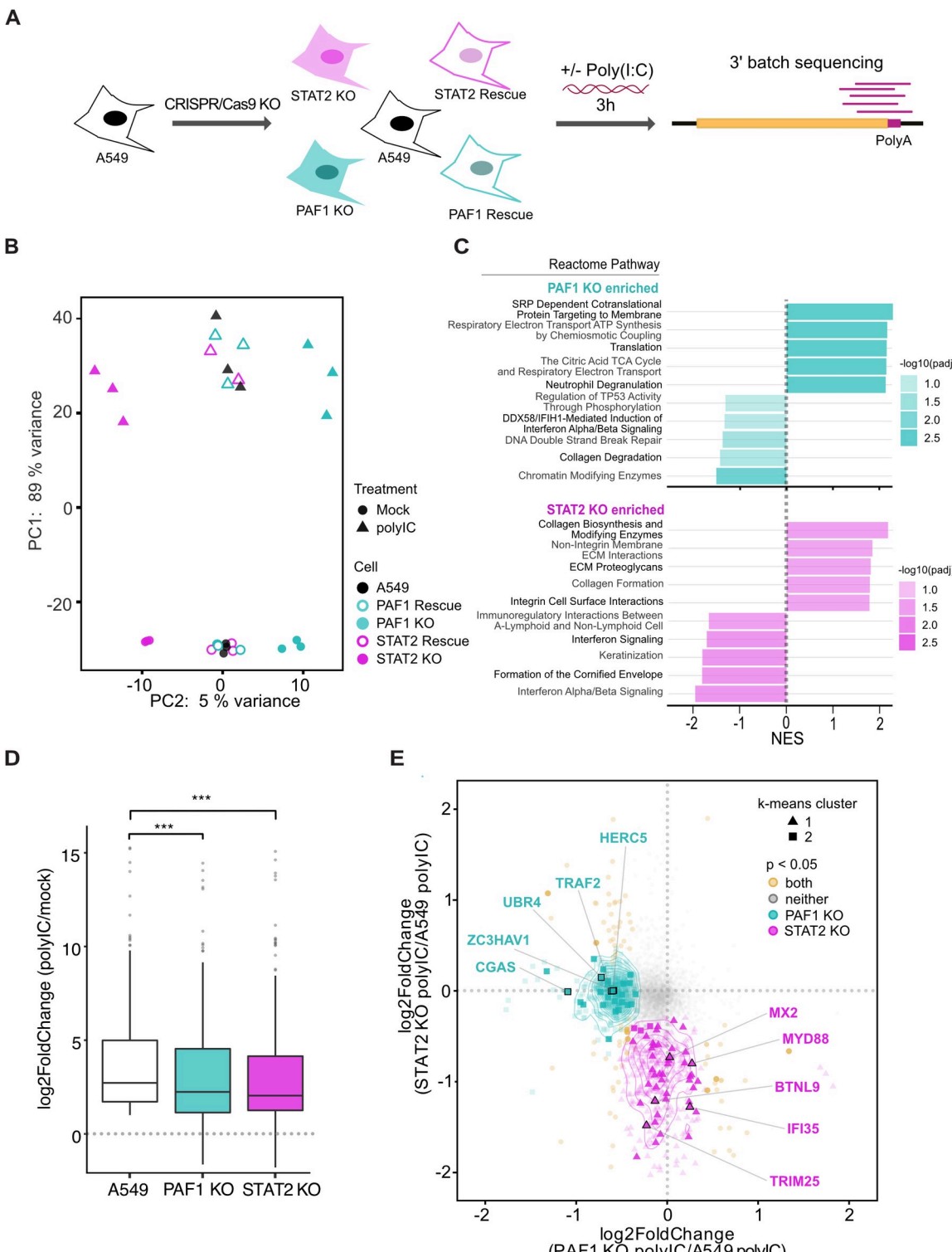

**Fig 2. PAF1 is required for expression of a subset of genes activated by poly(I:C).** (A) Parental A549, PAF1 KO/rescue and STAT2 KO/rescue cells were stimulated with poly(I:C) for 3 hours and subjected to RNA-seq and DESeq2 differential gene expression analysis. Results are based on three independent biological replicates. (B) Principal component analysis performed on all samples showed a clear separation between principal components (PC) describing untreated/treated cells and cell genotype. (C) GSEA was performed on genes differentially expressed in KO cells compared to parental A549 cells following poly(I:C) treatment. Up to the top 5 positively and negatively enriched Reactome pathways were plotted for each comparison (p adj < 0.1). A full list of GSEA results is available in S2

Table. (D) Changes in gene expression caused by poly(I:C) treatment are shown for the subset of immune response genes (GO:0006955) significantly upregulated for poly(I:C)-treated parental A549 cells relative to mock-treated A549 cells (log2 fold change > 0.5, padj < 0.05). A Wilcoxon signed rank test with Bonferroni correction was performed to identify significant changes caused by PAF1 or STAT2 KO. (E) All genes significantly upregulated for poly(I:C)-treated parental A549 cells relative to mock-treated A549 cells (log2 fold change > 0.5, padj < 0.05) were plotted based on log2 fold change of PAF1 and STAT2 KO cells relative to parental A549 cells following poly(I:C) treatment. Parental A549 cells were used as a normalization so that it was identical for both comparisons. Unsupervised K-means clustering was also performed to identify genes with similar behavior (triangles, circles and squares). P values were adjusted for false discovery rate using the Benjamini Hochberg method. Significant changes in gene expression are plotted for PAF1 KO (cyan), STAT2 KO (magenta), both (yellow) or neither (grey). Immune response genes (GO:0006955) are highlighted with larger markers and opaque coloring.

signaling. In yeast, the homolog of human PAF1C is associated with histone modifications that affect gene expression [30]. However, we did not observe any clear differences in global histone methylation in our PAF1 KO cells compared to parental A549 cells (S2 Fig), suggesting that these changes are not a result of global changes in PAF1 activity, but more gene specific.

Surprisingly, significantly upregulated pathways in PAF1 KO cells revealed several categories that may be pro-flaviviral in nature (Fig 2C). While generally thought of as a transcription elongation factor that promotes gene expression, PAF1C can repress expression in some contexts, such as genes with highly active super-enhancers in cancer cells [31]. Given the context-dependent nature of PAF1C in promoting or repressing gene expression, we hypothesized that PAF1C-mediated repression of pro-flaviviral genes may contribute to PAF1C restriction of DENV2 replication. As such, we examined genes composing the upregulated pathways in PAF1 KO cells and found many were related to known flavivirus host dependency factors. To rigorously test if PAF1 was repressing the expression of pro-flaviviral factors, we performed GSEA on flavivirus host dependency factors identified through genetic screens [32–37] (S3 Table). We found that PAF1 KO cells had significant upregulation of expression of these genes (p adj < 0.005) (S3 Fig). This effect is PAF1-specific as STAT2 KO cells did not show a significant change in expression for this group of genes.

We next explored the role of PAF1 in the expression of specific genes. We analyzed immune response genes (GO:0006955) whose expression was significantly induced in parental A549 cells following poly(I:C) treatment (log2 fold change > 0.5, p adj < 0.05) (S4 Table). This subset of genes allows the unbiased comparison of PAF1 and STAT2 KO cells to parental A549 cells in the context of the immune response. Both PAF1 and STAT2 KO resulted in the significant inhibition of expression of these immune response genes relative to parental A549 cells (Fig 2D). We also looked at genes that were significantly downregulated in PAF1 or STAT2 KO cells compared to A549 cells following poly(I:C) treatment (Fig 2E). While a few genes were significantly affected by both PAF1 and STAT2 KO, many genes displayed PAF1- and STAT2-specific responses. The overlap between the PAF1- and STAT2-dependent gene sets was significantly lower than expected by chance (p < 0.05). Unbiased k-means clustering formed unique clusters comprising PAF1- and STAT2-dependent genes, further underlining these differences. Our results indicate PAF1 and STAT2 have distinct but complementary roles and might function in synergy following poly(I:C) stimulation.

## Nuclear localization is required for the NS5-PAF1 interaction

To establish if the NS5 interaction with PAF1C is required for antagonism, we set out to identify the viral determinants of the interaction. Given the strong nuclear localization of NS5 during infection [8,10,38], we tested if this nuclear localization is required for its interaction with PAF1. We generated NS5 mutants for the central and C-terminal nuclear localization signals (NLSs) alone and together (Fig 3A) [8,9]. We used immunofluorescence confocal microscopy to determine the subcellular localization of NS5 mutants. Compared to WT NS5 (NS5$_{WT}$), the

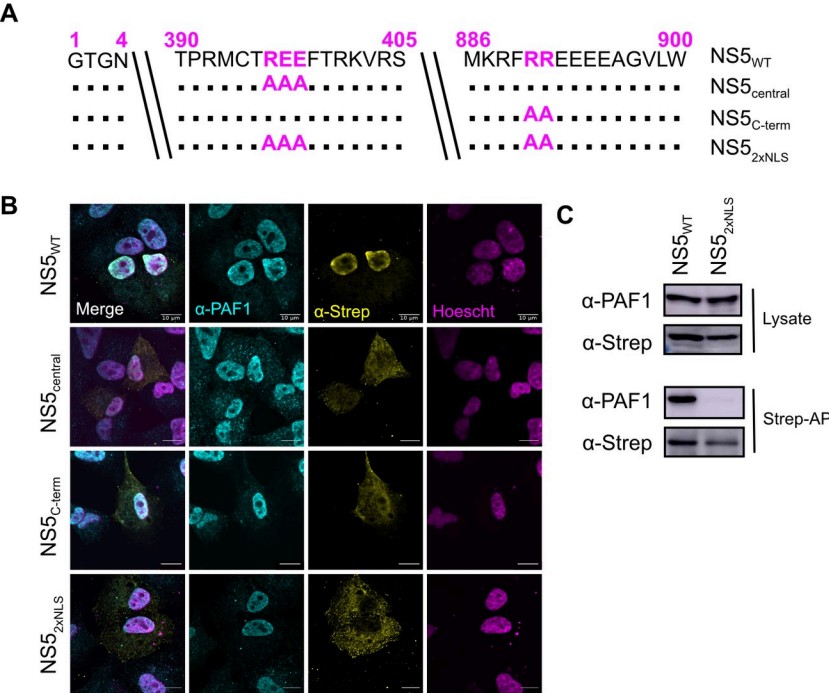

**Fig 3. Nuclear localization is required for NS5-PAF1C interaction.** (A) NLS mutants, including single mutant NS5$_{central}$ (REE396-398AAA) and NS5$_{C-term}$ (RR890-891AA) and the double mutant NS5$_{2xNLS}$ (REE396-398AAA and RR890-891AA). (B) Subcellular localization of 2xStrep II tagged NS5s (yellow) and colocalization with PAF1 (cyan) was determined by immunostaining and confocal microscopy. Nuclei were stained with Hoechst (magenta). Scale bar represents 10 μm. (C) The protein interaction between NS5$_{2xNLS}$ and PAF1 was tested biochemically. Plasmids encoding NS5s and GFP were transfected and affinity purified in HEK293T cells via a 2xStrep II tag. Immunoblot analysis of lysate and purified (Strep-AP) fractions was performed against PAF1, Strep and GAPDH (loading/negative control). Abbreviations: Strep-affinity purified (Strep-AP).

single NLS mutants (NS5$_{central}$ and NS5$_{C-term}$) showed some decrease in nuclear signal. Only the double mutant (NS5$_{2xNLS}$) fully abrogated nuclear localization and colocalization with PAF1 (Fig 3B).

We next determined if nuclear localization of NS5 is required for its interaction with PAF1C using affinity purification and immunoblot of NS5$_{WT}$ and the NS5$_{2xNLS}$ mutant. As expected, we observed an interaction between NS5$_{WT}$ and PAF1. This interaction was lost for NS5$_{2xNLS}$ (Fig 3C). Moreover, while we observed PAF1 and WT NS5 in cytoplasmic and nuclear compartments using biochemical fractionation, we only found NS5 to interact with PAF1 in the nuclear fraction (S4A Fig). There were several smaller molecular weight bands present in our immunoblots against the 2xStrep II tag, however most of these bands were also observed using an antibody against NS5 (S4B Fig), suggesting these smaller molecular weight bands are N-terminal degradation products that retain the C-terminal 2xStrep II tag. Together, our results show that NS5 nuclear localization is required for its interaction with PAF1.

## A highly conserved region of the C-terminus of the NS5 MTase domain is required for the interaction with PAF1C

While the nuclear localization of NS5 is required for the NS5-PAF1 interaction, the viral molecular determinants of the interaction surface may be different. To identify these determinants, we generated a series of N- and C-terminally truncated NS5 constructs (Fig 4A).

Importantly, all constructs conserved the central NLS region necessary for some nuclear localization and interaction with PAF1. The subcellular localization of all our truncations was assayed by immunofluorescence confocal microscopy (Fig 4B). As expected, all truncations had some nuclear localization. We also observed that truncations missing the C-terminal NLS (T4/5/6/7) were less abundant in the nucleus, reenforcing the importance of both the central and C-terminal NLS for nuclear localization [8,9].

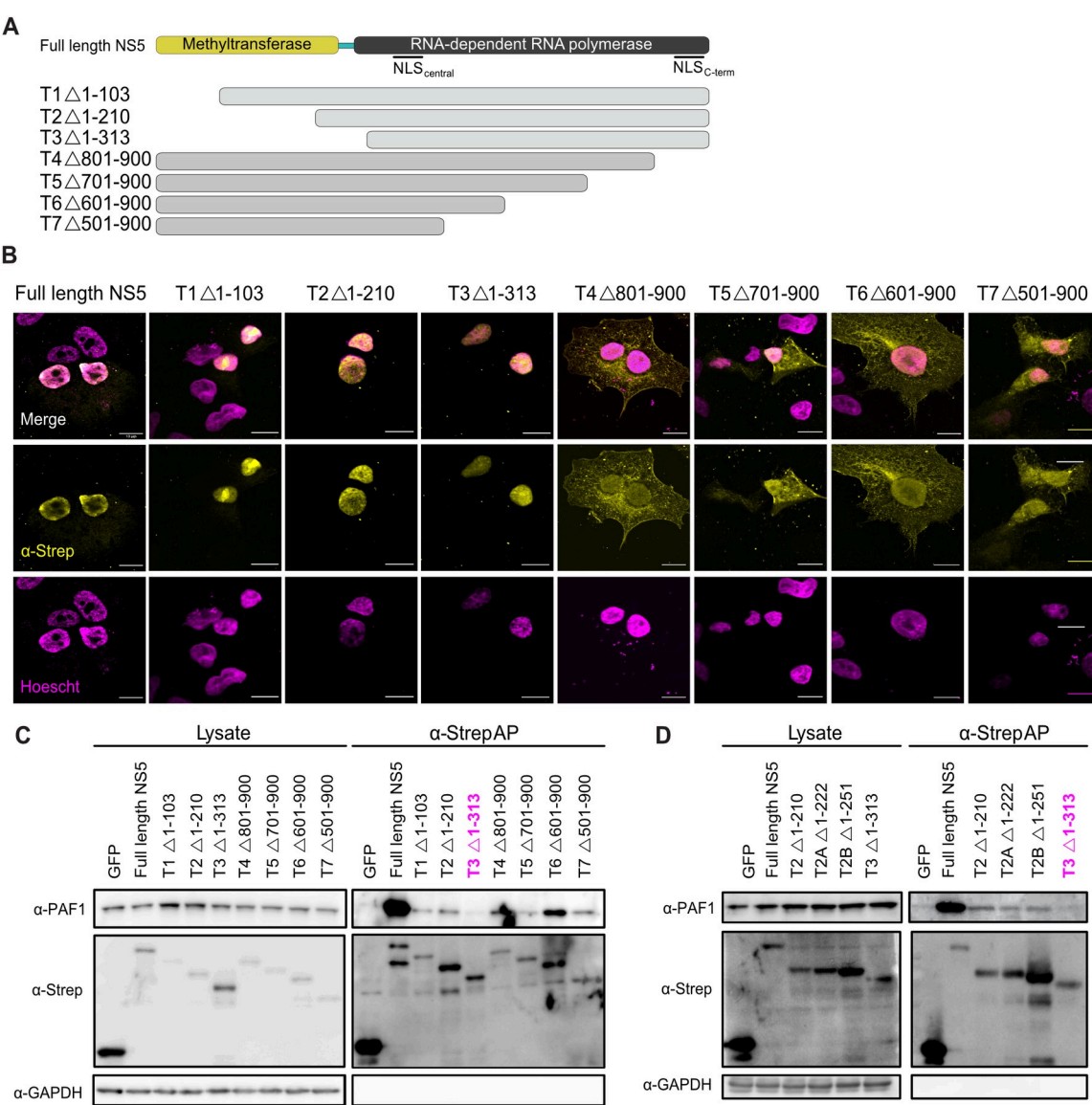

**Fig 4. A C-terminal region of MTase domain is responsible for interaction with PAF1C.** (A) NS5 truncations constructed to test NS5-PAF1 interaction. Each truncation corresponds to ~100 amino acid deletions from the N- (T1, T2, T3) or C-terminus (T4, T5, T6, T7) of NS5. (B) Subcellular localization of 2xStrep II tagged truncations and NS5$_{full-length}$ (yellow) and colocalization with PAF1 (cyan) were determined by immunostaining and confocal microscopy. Nuclei were stained with Hoechst (magenta). Scale bar represents 10 μm. (C) The protein interaction between NS5 truncations and PAF1 were tested biochemically. Plasmids encoding NS5 truncations, NS5$_{full-length}$ and GFP were transfected and affinity purified in HEK293T cells via a 2xStrep II tag. Immunoblot analysis of lysate and purified (Strep-AP) fractions were performed against PAF1, Strep and GAPDH (loading/negative control). (D) Refined truncations in the T2-T3 region (T2, T2A, T2B and T3) were tested for an interaction with PAF1 biochemically. Purification and immunoblot analysis were conducted as in (C). Abbreviations: Strep-affinity purified (Strep-AP).

To determine which region of NS5 interacts with PAF1, we used affinity purification and immunoblot. All truncations except T3 (Δ1–313) interacted with PAF1 (Fig 4C), meaning NS5-PAF1 interaction determinants are in the region separating T2 and T3 (amino acids 210–313). This region contains the C-terminus of the MTase domain, the linker and the N-terminus of the RdRp domain. We hypothesized that PAF1C interacts with the MTase region since PAF1C also interacts with host methyltransferases that use S-adenosyl-methionine as a methyl donor [39,40]. Therefore, we generated two new truncations to target this region, T2A (Δ1–222) and T2B (Δ1–251). Both truncations interacted with PAF1 (Fig 4D), suggesting that the PAF1-interacting region of NS5 resides between amino acids 251 and 265 of the MTase.

Conservation could help to identify specific amino acids of NS5 critical for its interaction with PAF1C. We and others previously observed the conservation of the NS5-PAF1C interaction across mosquito-borne flaviviruses [14,41]. We further sought to determine if this interaction was conserved amongst additional flaviviruses, including tick-borne flaviviruses. We tested different flavivirus NS5 proteins for subcellular localization and found that all had some nuclear localization, making an interaction with PAF1C possible (S5A Fig). Using affinity purification and immunoblot, we found all NS5s tested interacted with PAF1, including tick-borne encephalitis virus (TBEV), Langat virus (LGTV) and Saint-Louis Encephalitis virus (SLEV) (Fig 5A).

Alignment of the NS5 protein sequences between amino acids 251 and 265 revealed a highly conserved stretch of six amino acids that could be responsible for the interaction with PAF1C (Fig 5B). This region precedes the pivot region of the inter-domain linker important for NS5 folding [42,43] (Fig 5C). We performed site-directed mutagenesis on these six amino acids, creating two new NS5 mutants, LGS258AAA (NS5$_{LGS}$) and GTR261AAA (NS5$_{GTR}$) (Fig 5B). While the NS5 mutants localized predominantly in the nucleus (S5A Fig), they failed to interact with all four principal components of PAF1C (PAF1, LEO1, CDC73, and CTR9) (Fig 5D). These mutants still bound STAT2, albeit at a reduced level. Even with longer exposure times, the mutants did not show binding to PAF1 (S5B Fig). In summary, residues in the C-terminus of the NS5 MTase domain are required for the NS5-PAF1C interaction.

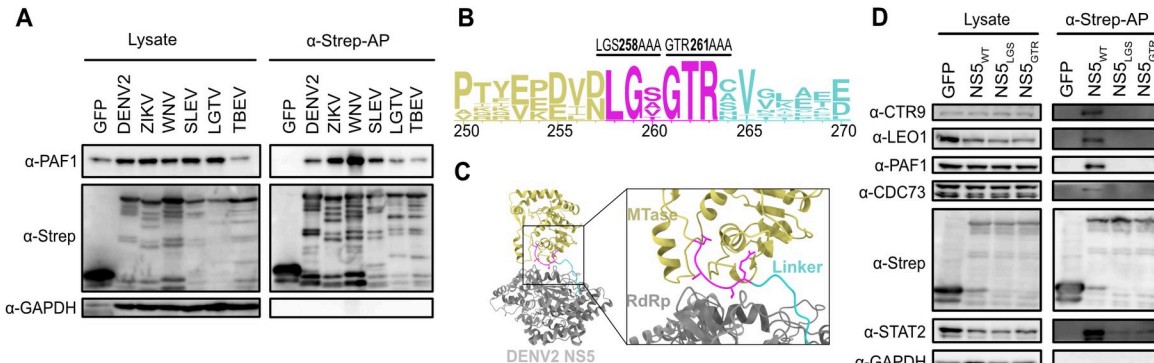

**Fig 5. A highly conserved region of the MTase C-terminus is required for PAF1C binding.** (A) The protein interaction between flavivirus NS5s and PAF1 was tested biochemically. Plasmids encoding NS5s and GFP were transfected and affinity purified in HEK293T cells via a 2xStrep II tag. Immunoblot analysis of lysate and purified (Strep-AP) fractions were performed against PAF1, Strep, and GAPDH (loading/negative control). (B) Logo analysis of amino acid conservation for flavivirus NS5s from (A) in the PAF1-interacting region. The conserved stretch from amino acids 258–263 (magenta) was targeted for alanine scanning leading to the generation of 2 NS5 mutants: LGS258AAA (NS5$_{LGS}$) and GTR261AAA (NS5$_{GTR}$). (C) DENV2 NS5 structure from PDB (5ZQK) with PAF1-interacting region highlighted (box). The PAF1-interacting region overlaps with the C-terminal end of the MTase (yellow) and the flexible linker domain (cyan), but not the RdRP (grey). The PAF1-interacting region includes a stretch of amino acids conserved in PAF1-interacting flaviviruses tested in (A). (D) Mutants from (B) were tested for an interaction with PAF1C biochemically. Purification and immunoblot analysis were conducted as in (A). PAF1C complex members CTR9, LEO1, CDC73 and PAF1 were probed. STAT2 was used as a positive control for an NS5 interaction outside of the PAF1C-interacting region. Abbreviations: Zika virus (ZIKV), West Nile virus (WNV), Powassan virus (POWV), Langat virus (LGTV), tick-borne encephalitis virus (TBEV), Strep-affinity purified (Strep-AP).

## The NS5-PAF1C interaction is required to antagonize PAF1-dependent expression of immune response genes

We next determined if breaking the NS5-PAF1C interaction affects NS5 antagonism of immune gene expression. A549 cells expressing different NS5s (NS5$_{WT}$, NS5$_{LGS}$, NS5$_{GTR}$, and NS5$_{2xNLS}$) were subjected to 3' batch-tag RNA-seq following poly(I:C) stimulation (Fig 6A). All NS5 constructs showed similar levels of expression, diminishing the likelihood that differences in gene expression are a result of lower overall expression of certain NS5 mutants (S6A Fig). Surprisingly, we observed significantly diminished induction of immune response genes in this experiment relative to poly(I:C) treated parental A549 cells, despite using the same concentration of poly(I:C) and timepoint (S6B Fig and S5 Table). In exploring why this induction was dampened, we found strong induction of immune response genes in our mock samples from plasmid transfection alone, likely due to the use of lipofectamine 3000 [44]. In fact, when comparing mock and poly(I:C)-treated plasmid-transfected cells, the induction of immune response genes in mock plasmid-transfected cells was more correlated with the induction of these same genes in A549 cells following poly(I:C) treatment (S6C Fig and S5 Table). We consequently focused our analysis on genes significantly induced (log2 fold change > 0.5, p adj 0.05) under these more correlated stimulations.

We explored general trends using GSEA on genes differentially expressed following transfection of WT versus mutant NS5. GSEA revealed several interesting trends (Fig 6B and S2 Table). We observed significant positive enrichment of PAF1-dependent genes for NS5$_{LGS}$ and NS5$_{2xNLS}$ compared to NS5$_{WT}$ (p adj < 0.1 and 0.05, respectively), indicating these mutants fail to antagonize PAF1 as effectively as NS5$_{WT}$. NS5 mutants also showed significant positive enrichment of genes related to IFN signaling compared to NS5$_{WT}$ (p adj < 0.05). Importantly, there was also significant positive enrichment for genes related to DDX58-IFIH1 (RIG-I-MDA5) mediated induction of IFN-I for all three mutants (p adj < 0.05). The same pathway was significantly downregulated in our PAF1 KO cells (Fig 2C), supporting our hypothesis that the NS5-PAF1C protein interaction antagonizes expression of PAF1-dependent genes in this signaling pathway to enhance replication.

We next explored if the specific genes driving these general similarities are common or unique for each mutant. Volcano plots showed NS5 mutants predominantly displayed significantly increased expression of genes relative to NS5$_{WT}$, rather than decreases (|log2 fold change| > 0.5, p adj < 0.05) (S7 Fig and S5 Table). By comparing the three NS5 mutants to each other using hierarchical clustering, we found that genes significantly perturbed by NS5$_{2xNLS}$ formed a distinct cluster from NS5$_{LGS}$ and NS5$_{GTR}$. These results suggest that the disruption of nuclear localization affects a broader set of genes than targeted disruption of the PAF1C interaction domain (Fig 6C). We further identified specific genes whose expression was rescued by all three mutants (*e.g.* IFIT2, DDX58), or specific to NS5$_{2xNLS}$ (*e.g.* CXCL10, IFNB1).

We finally sought to determine how gene expression rescued by NS5 mutants related to antagonism of PAF1. We compared the expression changes between mutant and WT NS5 to changes from PAF1 and STAT2 KO (Fig 7A and S1 and S5 Tables). NS5 mutants rescued the expression of many genes inhibited by NS5$_{WT}$, some of which were PAF1-dependent (p adj < 0.05). This subset of rescued PAF1-dependent genes was consistent between all three mutants, suggesting a crucial role for the NS5-PAF1C interaction in antagonizing the expression of these genes, specifically. These genes were also STAT2-independent (Fig 7B and S1 and S5 Tables). Notably, for the immune response genes whose expression was rescued by NS5 mutants, there was a shift towards PAF1-dependency but not STAT2-dependency. This shift was statistically significant for all mutants (p < 0.05) and further underlines the

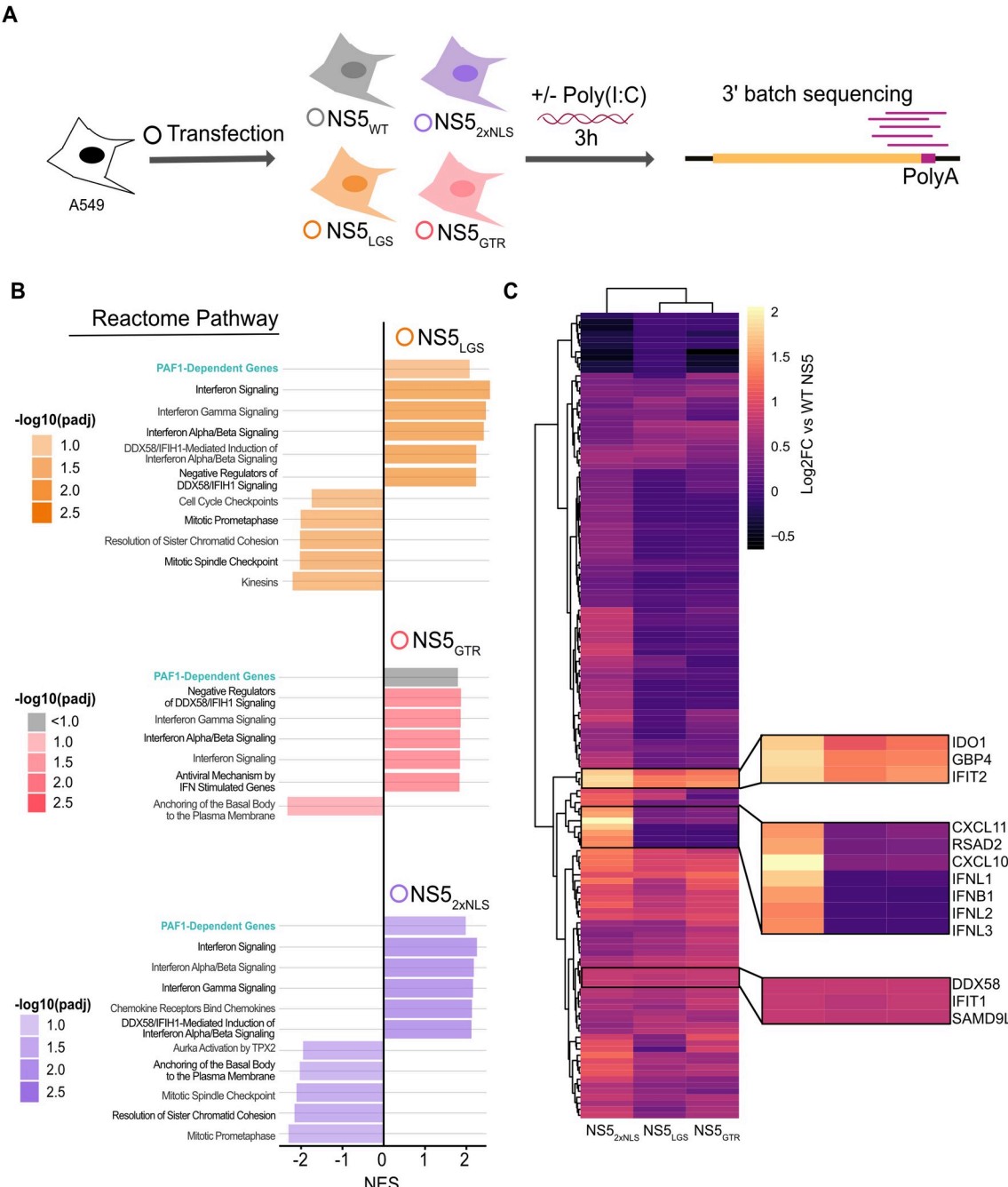

**Fig 6. NS5 mutants have shared and distinct impacts on gene expression.** (A) Parental A549 cells were transfected with NS5$_{WT}$ (grey), NS5$_{2xNLS}$ (purple), NS5$_{LGS}$ (orange), NS5$_{GTR}$ (red), or GFP as a negative control. Transfected cells were stimulated with poly(I: C) for three hours and subjected to RNA-seq. Results are based on three independent biological replicates. (B) GSEA was conducted on genes differentially expressed by cells expressing WT versus mutant NS5s. Up to the top 5 positively and negatively enriched Reactome pathways were plotted for each comparison (p adj < 0.1). A full list of GSEA results is available in S2 Table. (C) Genes significantly upregulated in NS5$_{2xNLS}$ versus WT NS5 (log2 fold change > 0.5, p adj < 0.05) were subjected to hierarchical clustering for each mutant and displayed as a heat map. Genes of interest are highlighted by zoom panels.

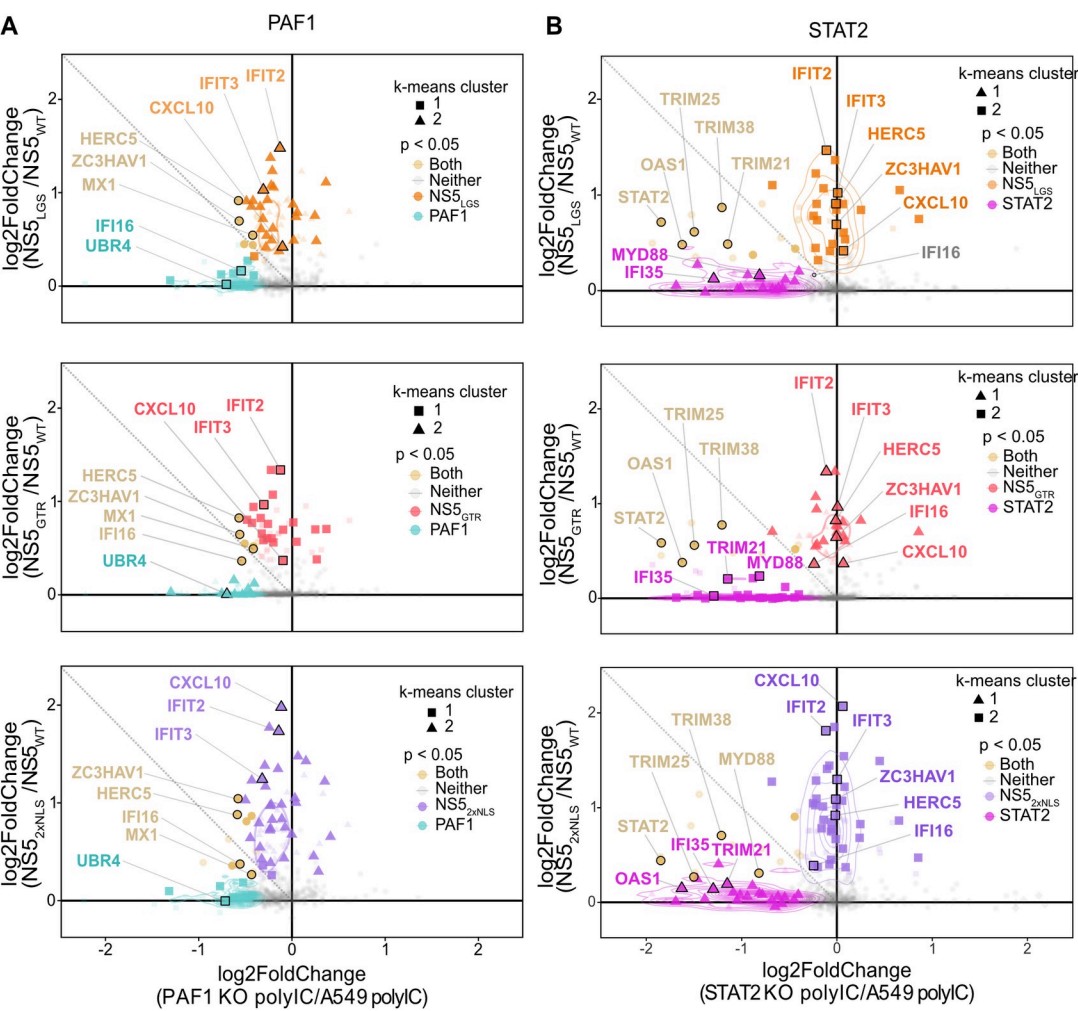

**Fig 7. NS5 mutants rescue expression of PAF1-dependent genes.** Relative gene expression was plotted as log2 fold change for (A) PAF1 KO versus parental A549 cells or (B) STAT2 KO versus parental A549 (same as Fig 2E), and NS5 mutant versus WT. Unsupervised K-means clustering was also performed to identify genes with similar behavior (triangles, circles and squares). P values were adjusted for false discovery rate using the Benjamini Hochberg method. Significant changes in gene expression are plotted for: PAF1 KO (cyan), both (yellow), neither (grey), NS5$_{LGS}$ (orange), NS5$_{GTR}$ (red) and NS5$_{2xNLS}$ (purple). Immune response genes (GO:0006955) are highlight with larger markers and opaque coloring.

complementary nature of NS5 antagonism of PAF1C in the nucleus. We further tested the subset of PAF1-dependent and STAT2-independent immune response genes using qRT-PCR (S8 Fig). Expression of these genes was rescued by NS5 mutants, though the effects were most consistent for the NS5$_{GTR}$ mutant. Thus, breaking the NS5-PAF1C interaction rescues the expression of PAF1-dependent genes. However, these mutants also rescue the expression of immune response genes that are PAF1- and STAT2-independent, suggesting additional mechanisms of immune antagonism by NS5.

## Discussion

Here, we explore gene expression mediated by PAF1C and antagonized by DENV2 NS5. We find that PAF1 restricts DENV2 infectious virion production and is required for the expression of poly(I:C)-induced genes that are STAT2-independent. We further map the viral

determinants of the NS5-PAF1C interaction and show that both NS5 nuclear localization and residues in the C-terminus of the MTase domain are necessary for this interaction. Finally, we show that breaking the NS5-PAF1C interaction rescues expression of a subset of PAF1-dependent and STAT2-independent immune response genes.

PAF1C has been viewed as a regulator of the host immune response and an antiviral host restriction factor [15]. We previously showed that knockdown of PAF1C members increased DENV2 and Zika virus (ZIKV) replication measured by immunostaining for E antigen [14]. Here, we extend these results to demonstrate that PAF1 restricts DENV2 infectious virion production (Fig 1). This stands in contrast to a recent study with Japanese encephalitis virus (JEV), in which JEV appears to depend on PAF1 for viral protein production to some degree [41]. Notably, JEV NS5 was also found to interact with PAF1C, suggesting that while the NS5-PAF1C interaction is broadly conserved among mosquito- and tick-borne flaviviruses (Fig 5A), the role of PAF1C in flavivirus replication may be virus or cell-type dependent.

Compared to previous studies, which focused on PAF1-mediated expression under basal conditions [45,46] or following IFN-I treatment [15], we identified PAF1-dependent immune response genes upstream of IFN-I signaling and/or independent of STAT2 (Fig 2). We also identified genes suppressed by PAF1, including flavivirus host dependency factors. Thus, PAF1 potentially restricts DENV2 replication through two independent mechanisms–upregulation of the immune response and suppression of proviral genes. PAF1 is known to have context-dependent impacts on transcription elongation in cancer cell lines [45,46] but appears to exclusively promote transcriptional elongation in primary mammalian cells [47]. Thus, studying the PAF1C-mediated immune response in primary cells will be an important future step.

The innate immune response serves as a cell's first response to pathogen infection, which flavivirus NS proteins antagonize. NS5 is a prominent inhibitor of IFN-I responses, although mechanisms of antagonism are virus-dependent [6,48–50]. Here, we demonstrate that NS5 interacts with PAF1C to inhibit the expression of PAF1-dependent immune response genes in a manner that is complementary to the established DENV NS5 antagonism of STAT2 [6]. In particular, genes related to the DDX58-IFIH1 (RIG-I-MDA5) signaling pathway were found to be PAF1-dependent and antagonized by the NS5-PAF1C interaction. This could be advantageous to DENV given the role of both RIG-I and MDA5 in restricting DENV replication [24–29].

Our work also revealed novel insights into NS5 antagonism of IFN-I signaling. Compared to NS5 mutants, NS5$_{WT}$ still inhibited genes related to IFN-I signaling despite not being able to degrade STAT2. Reduced binding of STAT2 by our NS5$_{GTR}$ and NS5$_{LGS}$ mutants may affect this process (Fig 5D). However, many of these genes are also STAT2-independent in A549 cells (Fig 7B). This could point to previously unknown mechanisms of NS5 antagonism of IFN-I signaling such as upstream DDX58-IFIH1 signaling. It will be valuable to study the effects of these mutations in the context of infection. Fortunately, mutations in this region have produced viable DENV2 mutants in the past [42].

Our studies reveal a novel role of nuclear NS5 in antagonizing the host immune response. Previous studies with infectious NS5 NLS mutants found no defects in overcoming IFN-I signaling [8]. However, mutants with complete loss of NS5 nuclear localization could not be studied using this approach because of their lethality, nor were mechanisms upstream of IFN-I signaling studied. Here, we show that nuclear localization of NS5 is important for inhibiting the expression of immune response genes, including those that are PAF1- and STAT2-independent. Thus, NS5 nuclear localization could be a general mechanism by which DENV inhibits immune gene expression, potentially through protein interactions with host gene expression machinery in the nucleus [13,14]. This also appears to be a general mechanism for other flavivirus NS5 proteins. Similar to our observations for DENV2 NS5, a recent study

shows that nuclear WNV NS5 down-regulates immune gene expression in HEK 293T cells
[51]. Nuclear ZIKV NS5 also shows this behavior in HEK 293T cells [52], though it may be cell
type-specific [53]. Duck Tembusu virus (TMUV) NS5 is unique in having no detectable
nuclear localization [54]. Consequently, TMUV NS5 makes an interesting model for gain-of-
function experiments related to nuclear localization. Surprisingly, adding a NLS to TMUV
NS5 led to slightly increased IFNβ production in duck embryonic fibroblasts, however other
immune response genes were not explored [55]. Taken together, there is growing evidence
that immune antagonism is a recurring feature of flavivirus NS5 proteins in the nucleus, but
may not be intrinsic to all nuclear NS5 proteins in all contexts. Systematic comparative studies
dissecting flavivirus NS5 immune antagonism under identical conditions will establish the
breadth of this phenomenon. These comparative studies would also establish cell type specific-
ity, gene specificity, and the molecular mechanisms through which they are inhibited.

Extensive studies of PAF1C in yeast and mammals have revealed multiple roles in regulation
of gene expression. Depending on cell type, and environment, PAF1C exhibits different func-
tions impacting transcription mechanisms, chromatin modifications, and post-transcriptional
events (reviewed in [30]). PAF1C impacts on antiviral gene expression has been shown to be
virus dependent [15,56], complicating our understanding of PAF1C role during DENV2 infec-
tion. Our previous study [14] shows that PAF1C regulation of immune genes during DENV2
infection is linked to PAF1C occupation of the promoter region of ISGs. With our current
study we show an unchanged level of global methylation in in PAF1 KO cells (S2 Fig). There-
fore, we hypothesize that PAF1C effects on chromatin are gene-specific, or another function of
PAF1C is modulated. PAF1C positively regulates RNA Pol II release at paused genes, including
immune response genes. Not all immune response genes are regulated in this manner. For
example, IRF3-induced gene expression is mainly due to *de novo* recruitment of Pol II [57].
Nonetheless, PAF1-dependent immune response genes identified in our study were classified as
having "high" RNA Pol II pausing indices [58,59]. Thus, NS5 antagonism of PAF1-dependent
immune gene expression may act through the positive regulation of PAF1C on RNA Pol II
release at paused genes. While not significant (p = 0.06), the difference observed between paus-
ing index of PAF1-dependent anti- and pro-flavivirus genes supports this hypothesis (S7 Table,
genes selected from [58]). The different mechanisms used by STAT2-IRF3 and PAF1C regard-
ing the anti-viral gene expression highlights the necessity for DENV2 to develop an indepen-
dent antagonism of STAT2 and PAF1C. Interrogating these molecular mechanisms will be an
important future direction in dissecting the role of PAF1C and its interaction with NS5.

## Conclusions

In summary, we interrogate the role of PAF1C in DENV2 replication, regulating immune
gene expression and NS5 antagonism of this process. By mapping the viral determinants of the
NS5-PAF1C protein interaction and creating NS5 mutants in which the NS5-PAF1C interac-
tion is broken, we rescued the expression of PAF1-dependent immune response genes. These
mutants also rescued the expression of immune response genes that are PAF1- and STAT2-in-
dependent. As such, NS5 antagonizes the expression of immune response genes using multiple
mechanisms. Exploring PAF1- and STAT2-independent mechanisms in the future will reveal
novel insights into the role of nuclear NS5.

## Materials and methods

### Cells and viruses

A549 (ATCC), HEK293T (gift of Sam Díaz-Muñoz) and Vero (ATCC) cell lines were main-
tained in Dulbecco's modified Eagle's medium (DMEM, Gibco ThermoFisher) supplemented

with 10% fetal bovine serum (FBS, Gibco ThermoFisher) at 37˚C. Cells were tested for myco-plasma monthly by PCR. Clonal A549 cells knockout (KO) for STAT2 and PAF1 were gener-ated by CRISPR/Cas9. The CRISPR guide RNA sequences are from the GeCKO v2 Human library [60]. The gRNA oligos were cloned into the lentiCRISPRv2 puro plasmid (Addgene #52961, [60]). The cloning product (3.5 µg) were transfected using calcium phosphate protocol into 293T cells with lentiviral packaging plasmids including 1.8 µg pMDLg/p-RRE, 1.25 µg pCMV-VSV-g and 1.5 µg pRSV-Rev [61]. After 36 hours lentivirus particles were collected and cell debris was removed by centrifugation and filtration through a 0.45 µm filter. The resulting lentiviral stocks were used to transduce parental A549 cells. Transduced cells were selected for puromycin resistance (10 µg/ml, ThermoFisher). PAF1 KO generation required a clonal selection by limiting dilution. PAF1 and STAT2 KO were verified by Sanger sequencing and immunoblotting using antibodies against PAF1 (1:1000, Bethyl Labs, A300-173A) or STAT2 (1:200, Santa Cruz Biotechnology) (S1 Fig). A549 PAF1 and STAT2 rescue cells were created by cloning PAF1 and STAT2 cDNA into pLenti6 plasmid (Addgene #89766, [62]). Lentiviral packaging was performed as described above. Clonal PAF1 KO and bulk STAT2 KO cell lines were transduced and selected for Blasticidin resistance (12 µg/ml, InvivoGen) to gen-erate PAF1 and STAT2 rescue cells. No additional rounds of clonal selection were performed for rescue. High titer stocks of DENV2 16681 strain were obtained by passage of a p0 stock in C6/36 cells [14].

## Plasmids

All DENV2 truncated protein-encoding plasmids were generated in the pcDNA4_TO with a C-terminal 2xStrep II affinity tag using PCR amplification and Gibson assembly. Flaviviruses NS5-encoding plasmids were previously generated (DENV2, ZIKV, WNV) [14] or synthesized for Gibson assembly (TBEV, POWV, LGTV) (Genewiz, Gene Universal). DENV2 NS5 point mutations were generated using site-directed mutagenesis of pcDNA4_TO DENV2 WT NS5 plasmid. LentiCRISPRv2 Puro (Addgene #52961, [60] encoding Cas9 and gRNA targeting PAF1 and STAT2 were generated following the Sanjana et al. protocol. Plasmid encoding PAF1 was kind gift from Nevan Krogan. PAF1 and STAT2 (Addgene #71451, [63]) were inserted into pLenti6 via Gibson assembly. Primer sequences used for the generation of all our constructs are available in S6 Table.

## Virus infection and plaque assay

PAF1 KO or rescue A549 cells were inoculated at a MOI of 0.1 and supernatants were removed at the indicate time and stored at -80˚C. Samples were subjected to 10-fold serial dilutions fol-lowed by incubation on a monolayer of Vero cells for two hours at 37˚C. Samples were removed and replaced with 0.8% methylcellulose (Sigma, MO512) in 1X DMEM supple-mented with 2% FBS and 1% pen/strep. Plaque assays were incubated for 8 days at 37˚C, then fixed with 4% formaldehyde and stained with 0.23% crystal violet to visualize plaques.

## Affinity purification and immunoblotting

HEK293T cells were seeded with a density of $4 \times 10^6$ cells in 10 cm dishes and transfected with 5 µg of Strep-tagged plasmid encoding the truncated NS5 or the Flaviviruses NS5 using Polyjet (1:3 ratio, Signagen). After 36 hours post-transfection lysis, affinity purification, and washes were performed using buffer containing, Tris-HCL pH 7.4 50 mM, NaCl 150 mM, EDTA 1 mM and 0.5% NP40, supplemented with protease inhibitor mini tablets and phosphatase inhibitor mini tablets (ThermoFisher Scientific). The REAP protocol [64] was used to separate nuclear and cytoplasmic fractions. All samples were treated similarly for affinity purification.

Pulldown of Strep-tagged protein was performed using Strep-tactin beads (IBA Lifescience) as previously described [14]. Protein samples were resuspended in 4:1 NuPage:TCEP loading buffer, heated for 10 min at 95°C. Both cell lysates and affinity purification fractions were separated by SDS-Page on 7.5% polyacrylamide gels. Proteins were transferred onto PDVF 0.2 μm membrane (Hybond P, GE Healthcare). The membranes were blocked in 5% non-fat milk in 0.1% TBS-tween, then incubated overnight at 4°C or 2 hours at room temperature with primary antibody diluted in 2.5% BSA, TBS-tween 0.1%. The primary antibodies were detected by secondary antibody anti-mouse or anti-rabbit conjugated to horseradish peroxidase (Southern Biotech, 1:10,000 dilution) for 1 hour at room temperature. The antibody-bound proteins were detected by incubating with Pierce ECL substrate or SuperSignal West Femto substrate (ThermoFisher) and visualize on an Amersham Imager 600 system (GE Healthcare). The following antibodies and dilutions were used: Strep 1:1000 (Qiagen, 34850), PAF1 1:1000 (Bethyl Labs #A300-173A), CDC73 1:500 (GeneTex, GTX110280), LEO1 1:1000 (Bethyl Labs #A300-173A), STAT2 1:200 (Santa-Cruz Biotechnology, sc-514193), GAPDH 1:1000 (Cell Signaling Technology, 14C10), Calnexin 1:1000 (Cell Signaling Technology, C5C9) Lamin A/C 1:1000 (Cell Signaling Technology, 4C11), H3K79 1:1000 (Abcam, Ab3594), H2B 1:1000 (Abcam, Ab1790), H3K4me3 1:1000 (Abcam, Ab8580), H3K27 1:1000 (Abcam, Ab6002) and H3K9 1:1000 (Abcam, Ab8898).

## Confocal immunofluorescence microscopy

A549 cells were fixed with 4% paraformaldehyde 24 hours post transfection. Cells were permeabilized with 0.25% Triton X-100 for 5 min and blocked with 5% goat serum (Sigma) in PBS. Coverslips (#1.5) were incubated for 1 hour with primary antibody followed by 1 hour with secondary antibody at room temperature. Nuclei were visualized with Hoechst (Invitrogen). Confocal microscopy was performed with confocal Olympus FV 1000 microscope using 60x/1.35 oil immersion objective. AlexaFluor 488, AlexaFluor 555, and Hoechst 33342 were detected using 488 nm, 543 nm and 405 nm, respectively, all at 1:1000 dilution. For the microscopy we used the following antibodies: PAF1 1:500 (Atlas antibodies, HPA043637), Strep 1:1000 (Qiagen, 34850). For quantification of PAF1 intensity in parental and PAF1 rescue cells, corrected nuclear fluorescence intensity was calculated using Fiji [65] by outlining the nuclei of 36 cells across 8 images for each condition. The integrated density of each nucleus was corrected by subtracting the background signal.

## Transcriptomic sample preparation

Parental, PAF1 KO/ PAF1 rescue, STAT2 KO/ STAT2 rescue A549 cells were plated at a density of $9 \times 10^5$ cells per well in 6-well plates. Cells were stimulated with poly(I:C) at a final concentration of 2 μg/mL using 4 μL Lipofectamine 2000 (Thermo Fisher). At three hours post-transfection, total cellular RNA was extracted using the RNeasy mini kit (Qiagen). For experiments involved GFP and NS5 expression, 2.5 and 5 μg of plasmid DNA was transfected using a ratio of 1:2 plasmid:Lipofectamine 3000 36 hours before poly(I:C) stimulation. RNA samples were prepared with the QuantSeq3' mRNA-Seq Library kit for Illumina (Lexogen GmbH) following standard protocol [66]. The resulting cDNA libraries were analyzed with NextSeq 500 sequencing run with 4 million reads per sample.

## RNA-seq data processing

Quality of all reads was evaluated using FastQC (v0.11.5). To remove the 3' read end poly-A stretches the reads were trimmed (bbduk, sourceforge.net/projects/bbmap/). All remaining sequences were mapped against the human reference genome build 38 with STAR (v2.5.2b)

[67]. HTseq (v0.6.1) was used to count all reads for each gene and set up a read count table [68]. Differential gene expression analyses were performed using the DESeq2 Bioconductor package (v1.30.1) [69]. The default "apelgm" shrinkage (v1.28.0) [70] set up was used for our analysis. Gene-set enrichment analysis (GSEA) was performed with the fgsea Bioconductor package [71], using Reactome gene sets downloaded from the Molecular Signatures Database [72].

## Quantitative reverse transcription-PCR (qRT-PCR)

Total RNA was isolated using RNeasy mini kit (Qiagen) column purification. cDNA was synthesized with iScript kit (BioRad). Real-time PCR was performed with iTaq SYBR Green premix (BioRad) and data were collected with LightCycler 480 (Roche). All Ct values were normalized to the expression values of GAPDH RNA and gene expression quantification were performed by the $2^{-\Delta\Delta Ct}$ method [73]. The primers sequences are provided in S6 Table.

## Statistical analysis

For plaque assay of virus replication data, p values were calculated using a paired, one-tailed Student's t-test. For differential gene expression analysis, significant changes in specific genes (p adj < 0.05) were identified after adjusting for false discovery rate using the Benjamini Hochberg method. For GSEA, all genes were included in the random walk, with 10,000 iterations occurring for gene-sets of size 50 to 1000. Significant changes in the expression of groups of genes were calculated using a Wilcoxon signed rank test with Bonferroni correction for multiple testing for paired samples and Wilcoxon rank test with Bonferroni correction for multiple testing for unpaired samples. Overlap between gene sets was tested using a Fishers exact test. For the distribution of PAF1 fluorescence intensity, a p value was calculated using an F-test. Pausing index was analyzed by Wilcoxon rank test.

## Supporting information

**S1 Fig. Generation of PAF1 and STAT2 KO A549 cells.** (A) PAF1 gRNA target region and Sanger sequencing of the CRISPR/Cas9 induced deletion. Immunoblotting was performed on protein extracted from parental A549, PAF1 KO and PAF1 rescue. Immunostaining with PAF1 antibody showed the protein depletion for the PAF1 KO cells. PAF1 rescue data showed the restoration of the PAF1 detection. GAPDH is the control for protein loading. PAF1 gRNA target region and Sanger sequencing of the CRISPR/Cas9 induced deletion. (B) STAT2 gRNA target region and Sanger sequencing of the CRISPR/Cas9 induced deletion. Immunoblotting was performed on protein extracted from parental A549, PAF1 KO/rescue, and STAT2 KO/ rescue. Immunostaining with STAT2 antibody showed the protein depletion for the STAT2 KO cells. STAT2 rescue data showed the restoration of the STAT2 detection. GAPDH is the control for protein loading. DENV2 replication in (C) PAF1 KO expressing GFP and PAF1 rescue cells and (D) PAF1 rescue and A549 + nt gRNA (non-targeting gRNA) + GFP, MOI 0.1. Data from three replicates are plotted as mean values +/- standard deviation. P values were calculated using a paired, one-tailed Student's t-test. (E) Heterogeneity of PAF1 expression (cyan) was determined by immunostaining and confocal microscopy. Nuclei were stained with Hoechst (magenta). Scale bar represents 10 μm. A representative image is shown. (F) Quantification of PAF1 nuclear signal intensity across 36 nuclei. P-value was calculated using a F-test. Abbreviations: plaque forming units (pfu), not statistically significant (ns). (TIF)

**S2 Fig. State of histone methylation in A549, PAF1 KO and PAF1 rescue cells.** Comparison of global methylation levels in parental A549, PAF1 KO and PAF1 rescue cells. Immunoblotting was performed on protein extracted from parental A549, PAF1 KO/rescue. Immunostaining with H3K9me3, H3K27me3, H3K4me3, H3K79me3 and H2B antibodies showed unchanged level of detection across the different cell lines.
(TIF)

**S3 Fig. Impact of PAF1 KO on flavivirus host dependency factors.** (A) GSEA was performed using list of flavivirus host dependency genes. (B) Leading edge of flavivirus host dependency factors from GSEA. Heatmap represents log2 fold change relative to parental A549 following poly(I:C) treatment.
(TIF)

**S4 Fig. Subcellular localization of the NS5-PAF1 protein interaction.** (A) Following nuclear/cytoplasmic fractionation, NS5 and GFP were subjected to affinity purification and immunoblot. Lamin and Calnexin served as controls for fraction purity, GAPDH is the control for protein loading. (B) Immunoblot were performed on parental A549 cells transfected with 2xStrep II tagged DENV2 NS5. Immunoblot was probed with Strep, NS5 and PAF1 antibody. Similar band pattern is observed for NS5 and Strep staining. GAPDH is the control for protein loading.
(TIF)

**S5 Fig. Subcellular localization of flavivirus NS5s and mutant NS5s.** (A) Subcellular localization of 2xStrep II tagged flavivirus NS5s, $NS5_{LGS}$ and $NS5_{GTR}$ (yellow) was determined by immunostaining and confocal microscopy. Nuclei were stained with Hoechst (magenta). Scale bar represents 10 μm. (B) 2xStrep II tagged NS5s ($NS5_{WT}$, $NS5_{LGS}$ and $NS5_{GTR}$) were tested for an interaction with PAF1C biochemically. Affinity purification and immunoblot analysis were conducted on protein extraction from parental A549 cells transfected with $NS5_{WT}$, $NS5_{LGS}$ or $NS5_{GTR}$. PAF1 antibody was used to identify the PAF1-NS5 interaction. Only $NS5_{WT}$ showed a band for PAF1 staining, at both short and long exposure (x5). GAPDH is the control for protein loading.
(TIF)

**S6 Fig. Characterization of immune response in DNA-transfected A549 cells.** (A) Comparison of NS5 expression in parental A549 transfected with $NS5_{WT}$, $NS5_{LGS}$, $NS5_{GTR}$ and $NS5_{2xNLS}$. Immunoblotting was performed on protein extracted from transfected parental A549. Immunostaining with Strep antibody detected an equal level of transfected NS5s for all constructs. GAPDH is a control for protein loading. (B) Changes in gene expression caused by poly(I:C) treatment are shown for the subset of immune response genes (GO:0006955) significantly upregulated for poly(I:C)-treated parental A549 cells relative to mock-treated A549 cells (log2 fold change > 0.5, padj < 0.05). (C) Pearson's correlation coefficients were calculated for differential gene expression comparing DNA transfection and poly(I:C) stimulation.
(TIF)

**S7 Fig. Gene expression analysis of NS5 mutants.** Relative change in gene expression was plotted as log2 fold change versus adjusted p value to identify general trends for (A) $NS5_{LGS}$, (B) $NS5_{GTR}$, and (C) $NS5_{2xNLS}$ compared to $NS5_{WT}$. Genes with significant increases (log2 fold change > 0.5, padj <0.05) for $NS5_{2xNLS}$ were used for heatmap analysis in Fig 6C.
(TIF)

**S8 Fig. qRT-PCR analysis of PAF1-dependent genes rescued by NS5 mutants.** qRT-PCR was performed on PAF1-dependent immune response genes from Fig 7A. Fold changes were

calculated using the ΔΔCt method and normalized to GAPDH as the house-keeping gene. GFP transfection was used as a positive control for poly(I:C) induction.
(TIF)

**S1 Table. Differential gene expression data from PAF1 KO, PAF1 rescue, STAT2 KO, STAT2 rescue and A549 parental cell lines treated with poly(I:C).**
(XLSX)

**S2 Table. GSEA analysis results.**
(XLSX)

**S3 Table. List of pro-flaviviral genes.**
(XLSX)

**S4 Table. Summary of PAF1- and STAT2-dependent genes following poly(I:C) treatment.**
(XLSX)

**S5 Table. Differential gene expression data from A549 parental cell lines transfected with pcDNA_2xStrep- NS5 WT, NS5 $_{2xNLS}$, NS5 $_{LGS}$ or NS5 $_{GTR}$ and treated with poly(I:C).**
(XLSX)

**S6 Table. Oligonucleotides primers.**
(XLSX)

**S7 Table. Pausing index analysis of PAF1-dependent genes.**
(XLSX)

## Acknowledgments

We would like to thank members of the Shah lab, Ben Montpetit, and Marty Privalsky for fruitful discussions and critical feedback. This work is dedicated to the memory of Marty Privalsky, a wonderful colleague, scientist, and friend.

## Author Contributions

**Conceptualization:** Marine J. Petit, Matthew W. Kenaston, Priya S. Shah.

**Data curation:** Marine J. Petit, Matthew W. Kenaston, Ariana A. Nagainis, Priya S. Shah.

**Formal analysis:** Marine J. Petit, Matthew W. Kenaston, Oanh H. Pham, Ariana A. Nagainis, Adam T. Fishburn, Priya S. Shah.

**Funding acquisition:** Priya S. Shah.

**Investigation:** Marine J. Petit, Matthew W. Kenaston, Oanh H. Pham, Ariana A. Nagainis, Adam T. Fishburn, Priya S. Shah.

**Methodology:** Marine J. Petit, Matthew W. Kenaston, Oanh H. Pham, Ariana A. Nagainis, Adam T. Fishburn, Priya S. Shah.

**Project administration:** Priya S. Shah.

**Resources:** Priya S. Shah.

**Software:** Marine J. Petit, Matthew W. Kenaston, Priya S. Shah.

**Supervision:** Marine J. Petit, Priya S. Shah.

**Validation:** Marine J. Petit, Oanh H. Pham, Adam T. Fishburn.

**Visualization:** Marine J. Petit, Matthew W. Kenaston, Oanh H. Pham, Adam T. Fishburn, Priya S. Shah.

**Writing – original draft:** Marine J. Petit, Matthew W. Kenaston, Oanh H. Pham, Priya S. Shah.

**Writing – review & editing:** Marine J. Petit, Matthew W. Kenaston, Oanh H. Pham, Adam T. Fishburn, Priya S. Shah.

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
