## [Decision Letter · Decision Letter 0]

21 Jun 2021

Dear Dr. Shah,

Thank you very much for submitting your manuscript "Nuclear dengue virus NS5 antagonizes expression of PAF1-dependent immune response genes" for consideration at PLOS Pathogens. As with all papers reviewed by the journal, your manuscript was reviewed by members of the editorial board and by several independent reviewers. In light of the reviews (below this email), we would like to invite the resubmission of a revised version that takes into account the reviewers' comments. Specifically, it will be important to address how disruption of the NS5-PAF1 interaction by mutation of DENV affects viral replication and gene expression.

We cannot make any decision about publication until we have seen the revised manuscript and your response to the reviewers' comments. Your revised manuscript is also likely to be sent to reviewers for further evaluation.

Sincerely,

Stacy M Horner

Associate Editor

PLOS Pathogens

Christopher Basler

Section Editor

PLOS Pathogens

Kasturi Haldar

Editor-in-Chief

PLOS Pathogens

orcid.org/0000-0001-5065-158X

Michael Malim

Editor-in-Chief

PLOS Pathogens

orcid.org/0000-0002-7699-2064

Reviewer's Responses to Questions

**Part I - Summary**

Reviewer #1: This study extends previous research from the Shah lab investigating the role of the transcriptional PAF complex in the flavivirus lifecycle. In this study the utilise KO cells lines to indicate an effect on innate immune gene transcription that would normally aid the infected cell in controlling the infection. They also show that a conserved domain in the MTase region of NS5 determines a direct interaction with PAF1. Overall this is a very clear investigation that incorporates different approaches, but primarily protein mutation and expression and RNAseq to investigate their aims

Reviewer #2: This study builds on previous work that shows that the DENV2 protein NS5 can antagonize PAF1C recruitment to immune genes. Petit et al. demonstrate that PAF1C can contribute to restriction of DENV2 replication and provide evidence that PAF1 controls a transcriptional regulon independent from that of STAT2 following polyI:C treatment. By generating truncation mutants of NS5, they identify the viral determinants for interaction with PAF1C and show that disruption of these interactions limits NS5’s ability to control the PAF1 regulon. Their finding that NS5 can regulate expression of host genes independently of its role in STAT2 turnover is novel and provides new insight into how viral proteins modulate host gene expression.

The study is interesting and the biochemical experiments to map the NS5/PAF1 interaction domain are strong. However, several key experiments lack important controls and the authors fail to connect phenotypes characterized in polyI:C treated cells to cells infected with actual DENV2. Also, because the majority of the work relies on ectopic protein expression, the authors need to be more transparent about protein levels in various cell lines. If these weaknesses are addressed, this work will stand as a noteworthy contribution to our understanding of DENV-host interactions.

Reviewer #3: A number of papers, including from this group, have recently been published that have started to shed light on the nuclear functions of flavivirus NS5. Nuclear roles of NS5, the primary functions of which relate to viral replication, have long been a mystery since no stage of the flaviviral lifecycle involves the nucleus. Here, Petit et al. build on their earlier working showing that DENV NS5 targets PAF1C to inhibit immune genes by separating this PAF1C-dependent immune antagonism from the well-established function of NS5 in antagonising STAT2. Using CRISPR knockout cell lines, the authors show that PAF1C and STAT2 regulate a distinct set of immune genes, and that the molecular determinants required for NS5 antagonism of PAF1C can be disentangled from the molecular determinants of STAT2 antagonism. The authors also confirm that PAF1C is important for inhibiting DENV replication in an infection context.

The manuscript is well written and clearly presented and makes a valuable contribution to the field.

**Part II – Major Issues: Key Experiments Required for Acceptance**

Reviewer #1: - Fig 5A indicates that many flavivirus NS5 proteins interact with PAF1, but the immunoblots show an incredible multitude of bands of differing molecular weights. Could the authors provide some insight as to how they be sure of specificity if there are some many bands detected by the antibody? In all of the other assays shown which species is interacting with PAF1 directly then? Can this membrane be reprobed with anti-NS5 antibodies?

- are the authors able to determine if there has been any genome methylation induced by presence or absence of PAF1? This may be important in determining the transcription regulation and ultimate mechanism

Reviewer #2: 1. The viral replication experiment in Figure 1 lacks appropriate control cell lines. First, there is no wild-type equivalent cell line included for comparison. Because lentiviral transduction and clonal selection can alter cell responses, it is understandable that the authors did not compare their PFU data to a parental A549 cell. However, they need to compare their PAF1 KO cell line to a transduced/Cas9-ed cell in which PAF1 has not been manipulated. A scramble or untargeted gRNA control would work. A similar control is needed for the rescue cells (e.g. expression of a control protein like GFP in the KO background). It is definitely encouraging that the PAF1 and STAT2 rescues act similarly to WT A549s following polyI:C treatment (by virtue of clustering nicely in the Figure 2B PCA) but it is still important to show that the rescue phenotype is not being influenced by double transduction/selection and/or overexpression of an exogenous protein in the context of dengue virus infection.

Also relevant to Figure 1, there’s a question of how PAF1C protein expression in the rescue cell line compares to that in a WT A549 cell line. Is it possible to show a western blot comparing the PAF1C rescues to WT A549s (or better yet, the control cell lines requested above?)

2. The real “slam dunk” experiment that’s missing here would be to show that reintroducing PAF1-interaction mutant versions of NS5 into DENV2 impacts the expression of PAF1-dependent genes in the context of infection. While polyI:C is a reasonable proxy for DENV2-driven innate immune activation, the dynamics and levels of innate immune gene induction in a polyI:C transfection are vastly different from that of a bone fide viral infection. Likewise, NS5 levels are surely different in an overexpression cell line vs. a DENV2-infected cell. A set of parallel experiments to those in Figure 6, but with WT DENV2 and mutant DENV2, would significantly strengthen the manuscript’s conclusions.

Reviewer #3: Separation of the STAT2 and PAF1 binding domains of NS5 is critical for the interpretation of the data in the manuscript. In Fig 5D, STAT2 binding to the NS5 mutants is dramatically reduced (but still present). While binding of all PAF1C components to NS5 looks to be completely abrogated in the same experiment, the levels of binding of wild-type NS5 to PAF1C components are also lower than for STAT2. The distinction between abrogated PAF1C binding and reduced (but still present) STAT2 binding may therefore simply result from exposure differences in the western blot. It would be helpful to see a longer exposure of the PAF1C blots to show more convincingly that binding of PAF1C components are indeed completely abrogated. Quantification of binding levels would also be helpful in this regard.

**Part III – Minor Issues: Editorial and Data Presentation Modifications**

Reviewer #1: - the citation list is focussed primarily on Dengue reports but perhaps could be extended to other reports where RNA-seq has been used to investigate flavivirus infection and NS5 expression, particularly as the authors are indicating a potential conserved role here

Reviewer #2: Minor comments:

Line 48: unclear what methyltransferase activity is being referenced here.

Line 91: the meaning of this sentence is ambiguous. Is “the measurement of antigen-positive cells” the way in which you experimentally showed that PAF1C restricts DENV2 replication or is that the mechanism through which PAF1C restricts DENV2 replication?

PAF1C’s role in controlling RNAPII pausing and transcriptional elongation has been studied extensively but there is very little mention of it here. Could the authors elaborate in the discussion on the mechanistic role of PAF1C in the innate immune response and speculate as to why certain genes like HERC5, TRAF2, CGAS, etc. would be sensitive to control by PAF1C?

Figure 4 figure legend is confusing because the MTase domain itself is at the N-terminus of the protein. Perhaps “A C-terminal portion of the NS5 MTase domain is responsible for interaction with PAF1”

Additional detail is needed in the Results and/or methods sections as to how PAF1 KO and rescue cell line selections were carried out (e.g. Were the rescue cells made from clonal KOs? Did the rescue cells undergo multiple rounds of cloning?). If the rescues are not clonal, could we see an IF image showing how uniform expression of PAF1 is across multiple cells in the population?

Could the authors comment on the nature of the lower molecular weight bands in the anti-Strep blots in Figure 5A and 5B?

If I’m understanding this correctly, Figure 6 experiments look at parental A549 cells that have been transfected with different NS5 mutants. Are these proteins tagged? Can they be detected by western blot (against the tag, if they are tagged, or with an anti-NS5 antibody if they are not). Differences in expression of the proteins could impact the ability of these proteins to regulate gene expression in important ways and needs to be controlled for.

Language in line 227 is confusing. What knockout cells are you referring to?

Reviewer #3: 1. What are the levels of PAF1 in wild-type AF549 cells (Fig 1)? The authors only show PAF1 levels in PAF1 rescue cells, but the STAT2 rescue cells have lower STAT2 levels compared to wild-type AF549 cells (Fig S1). Gene expression and DENV replication is only shown in PAF1 knock-out cells relative to rescue cells, so it would be important to know how PAF1 expression compares in wild-type and rescue cells.

2. A loading control should be shown for the STAT2 western blot in Fig S1.

3. When the authors refer to the nuclear localisation of NS5 (throughout the manuscript), it could be clarified that this refers to steady-state nuclear localisation. NS5 from different DENV serotypes/strains can appear more or less nuclear at steady state during infection, but likely all NS5 shuttles in and out of the nucleus albeit with a different steady state equilibrium. This could also be introduced better in the paragraph starting on line 56.

4. The LGS/GTR mutants could be better explained in the legend to Fig 5.

PLOS authors have the option to publish the peer review history of their article (what does this mean?). If published, this will include your full peer review and any attached files.

Reviewer #1: **Yes: **Jason Mackenzie

Reviewer #2: No

Reviewer #3: No
---

## [Editor Report · Decision Letter 1]

8 Nov 2021

Dear Dr. Shah,

We are pleased to inform you that your manuscript 'Nuclear dengue virus NS5 antagonizes expression of PAF1-dependent immune response genes' has been provisionally accepted for publication in PLOS Pathogens.

Best regards,

Stacy M Horner

Associate Editor

PLOS Pathogens

Christopher Basler

Section Editor

PLOS Pathogens

Kasturi Haldar

Editor-in-Chief

PLOS Pathogens

orcid.org/0000-0001-5065-158X

Michael Malim

Editor-in-Chief

PLOS Pathogens

orcid.org/0000-0002-7699-2064
---

## [Editor Report · Acceptance letter]

15 Nov 2021

Dear Dr. Shah,

We are delighted to inform you that your manuscript, "Nuclear dengue virus NS5 antagonizes expression of PAF1-dependent immune response genes," has been formally accepted for publication in PLOS Pathogens.

Best regards,

Kasturi Haldar

Editor-in-Chief

PLOS Pathogens

orcid.org/0000-0001-5065-158X

Michael Malim

Editor-in-Chief

PLOS Pathogens

orcid.org/0000-0002-7699-2064